# Topical Treatment of Elevated Intraocular Pressure in Patients with Graves’ Orbitopathy

**DOI:** 10.3390/ijerph17249331

**Published:** 2020-12-13

**Authors:** Magdalena Gumińska, Roman Goś, Janusz Śmigielski, Michał S. Nowak

**Affiliations:** 1Provisus Eye Clinic, 112 Redzinska str., 42-209 Częstochowa, Poland; mguminska@op.pl; 2Department of Ophthalmology and Visual Rehabilitation, Central Veterans Hospital, Medical University of Lodz, 113 Zeromskiego str., 90-549 Lodz, Poland; rgos@poczta.onet.pl; 3Department of Statistics, State University of Applied Science in Konin, 1 Przyjazni str., 65-510 Konin, Poland; janusz.smigielski.stat@gmail.com; 4Saint Family Hospital Medical Center, 19 Wigury str., 90-302 Lodz, Poland

**Keywords:** Graves’ orbitopathy, exophthalmos, intraocular pressure, topical medications

## Abstract

*Purpose*: In this study, we evaluated the efficacy of topical hypotensive treatment and/or systemic corticosteroids therapy in patients with elevated intraocular pressure and Graves’ orbitopathy (GO). *Methods*: We included 172 eyes in 86 individuals with duration of GO ≥ 3 months, intraocular pressure in either eye ≥ 25.0 mmHg, and GO ranked ≥ 3 at least in one eye in modified CAS form. The study subjects were divided into three treatment subgroups: subgroup I was administered latanoprost once a day; subgroup II was administered a combined preparation of brimonidine and timolol BID; subgroup III was the control group, not receiving any topical hypotensive treatment. All the study participants received systemic treatment, intravenous corticosteroid therapy at the same dose, according to the European Group of Graves’ Orbitopathy (EUGOGO) guideline. *Results*: On the final visit, the mean IOP value was significantly lower in all treatment subgroups compared to the initial values. In both subgroups receiving topical treatment, the IOP reduction was higher than in the control group receiving systemic corticosteroids only. However, the latanoprost eye drops decreased intraocular pressure more effectively than drops containing brimonidine and timolol. *Conclusion*: Topical ocular hypotensive treatment is effective in reducing intraocular pressure in GO and decreases intraocular pressure more effectively than systemic corticosteroid therapy alone.

## 1. Introduction

Thyroid-associated orbitopathy, also called Graves’ orbitopathy (GO), is an autoimmune, inflammatory disease of the orbital tissue. It is caused by autoantibodies against the thyrotropin receptor on endothelial cells of the thyroid follicles and against a subpopulation of orbital fibroblasts. It occurs in about 25–50% of patients with Graves’ disease (GD) [1,2,3,4,5]. It is estimated that elevated intraocular pressure (IOP) in the course of GO affects 3.7–24% of patients. It is also known that, in most cases, it stabilizes following immunosuppressive therapy. However, in clinical practice, persisting increases in IOP, despite general treatment, are often observed [1,6,7,8,9,10,11]. 

Elevated IOP present in GO is a significant sign, not included in the disease severity classification. In consequence, it may result in glaucomatous optic neuropathy because (according to the European Glaucoma Society guidelines) ocular hypertension is the most important risk factor for the development of glaucoma [12]. The IOP is determined by aqueous humour production, aqueous humour outflow, and the level of episcleral venous pressure. In GO, increased orbital pressure results in increased episcleral venous pressure leading to elevated IOP [13]. According to various authors, glaucoma may occur in the course of GO in 0.8–13% of patients [1,6,7,8,10,11] and the number of studies concerning elevated IOP treatment in patients with GO is limited [13,14,15,16,17]. 

In the treatment of the elevated IOP in GO patients, routine topical hypotensive therapy can be successfully applied as for all patients with glaucoma or ocular hypertension [17]; the ideal choice of hypotensive drug seems to be important. Due to GO resulting in inflammation of the ocular surface, the ideal hypotensive drug should combine the best intraocular pressure-lowering effect with the best tolerance to ensure the patient’s compliance [18,19,20]. 

A prostaglandin-group drug (prostaglandin F2α analogue, latanoprost) and a complex drug, being a combination of brimonidine (clonidine derivative, an α_2_-selective agonist) and timolol (nonselective β receptor antagonist), were selected for our study. Prostaglandin-group drugs (i.e., latanoprost) increase the outflow of aqueous humour and a complex drug, being a combination of brimonidine and timolol (i.e., Combigan), decreases the production and simultaneously increases the outflow of aqueous humour. Latanoprost and Combigan showed good efficacy in decreasing IOP and a good ocular-tolerability profile in patients with glaucoma or ocular hypertension in many clinical studies [18]. The systemic corticosteroids therapy decreases the activity of the tissue inflammation in GO patients in both genomic and non-genomic pathways. However, patients treated with systemic corticosteroids should first be screened for liver dysfunction, uncontrolled hypertension, and/or diabetes, history of peptic ulcer, and glaucoma, and then monitored for possible adverse events [21,22,23].

The aim of the present study was the evaluation of the efficacy of elevated IOP treatment with the use of topical therapy with a prostaglandin-group drug and a complex drug, being a combination of a α2-mimetic and a β-blocker, and/or systemic corticosteroids therapy applied according to the European Group of Graves’ Orbitopathy (EUGOGO) guidelines in patients with orbitopathy in the course of Graves’ disease.

## 2. Materials and Methods 

A prospective study was conducted at the Department of Ophthalmology and Vision Rehabilitation, Central Veterans Hospital, Medical University of Lodz, in the period between June 2011 and June 2013. The study, performed within the doctoral thesis, included 172 eyes in 86 individuals, 25 men and 61 women, patients of the Department of Endocrinology, Medical University of Lodz, with GO. The Clinical Activity Scale (CAS) by Mourits and Weetmann, in an individually modified version, was used for the assessment of GO activity. All participants were interviewed and information, including brief details of medical history and the eye conditions, as well as demographic data, was collected. Comprehensive ophthalmic examination included: distance visual acuity (VA) testing, a cover test, binocular and color vision assessments, exophthalmos (EXO) measurements with Hertel exophthalmometer, intraocular pressure (IOP) measurements with the Goldman applanation tonometry, ultrasound pachymetry, slit lamp and indirect ophthalmoscopic evaluation of the anterior and posterior segments, and other examinations, where needed. Inclusion criteria were: person aged ≥18 years, duration of orbitopathy at least 3 months, intraocular pressure in either eye equal to or exceeding 25.0 mmHg, and ocular Graves’ orbitopathy in modified CAS form ranked ≥3 for at least in one eye. The exclusion criteria were: pregnancy or lactation period, topical hypotensive treatment prior to the study, closed or narrow angle glaucoma, pseudoexfoliation syndrome (PEX), rubeosis iridis, intraocular surgery in the study eye within 3 months prior to screening, medical history of any uveitis and ocular trauma, uncontrolled heart disease, severe respiratory syndrome, liver failure, and known allergy to prostaglandin-group drugs or α2-mimetic and β-blocker drugs. The study subjects were randomly placed into three treatment subgroups: subgroup I—20 patients (7 men and 13 women) with GO qualified for the study were administered a prostaglandin, latanoprost (0.05 mg/mL), taken once daily, in the evening, same time, at 8 p.m.; subgroup II—20 patients (6 men and 14 women) with GO qualified for the study were administered a combined preparation of 2 mg/mL brimonidine and timolol + 5 mg/mL Combigan taken twice daily, at the same time, at 9:00 a.m. and 4 p.m.; subgroup III—46 patients (12 men and 34 women) with GO and intraocular pressure equal to or exceeding 25.0 mmHg were qualified for the control group, neither received topical hypotensive treatment nor placebo eye drops. All the study participants received systemic treatment, intravenous corticosteroid therapy at the same dose, according to the EUGOGO guidelines, i.e., methylprednisolone 0.5 mg weekly for the first six weeks and then methylprednisolone 0.25 mg for the next six weeks. The observation period of the patients lasted 12 weeks.

The study was approved by the institutional review board of the Medical University of Lodz (Ethical Approval Code RNN/488/11/KB) and informed consent was obtained from all included subjects. All participants were counseled about the prognosis for their condition and the nature and possible consequences of the treatment were explained. For statistical analysis, the demographic data were anonymously recorded, and all procedures used adhered to the tenets of the Declaration of Helsinki. 

### Data Management and Statistical Analysis

Data were entered into the Microsoft Excel database and commercially available software STATISTICA v. 10.1 PL (StatSoft Polska, Krakow, Poland) was used to perform all statistical analyses. The statistical analyses included demographic data as well as medical history data and patients’ tests results obtained on both the screening and the final visits. The sex distribution was explored by the Chi squared (χ^2^) test. Other non-parametric methods based on ranks were used for the analyses of participants’ ages, body mass indexes (BMIs), GO duration times, CAS inflammation values, EXO values, and IOP measurements values. The Kruskal–Wallis analysis of variance was used to examine the differences between all treatment subgroups. The comparison between particular subgroups was performed using the Mann–Whitney U-test and Dunn’s test because of the Bonferroni correction. Multiple logistic regression models were used to investigate the association of sex and the duration of GO with EXO values on the screening visit. Differences were considered significant at *p* < 0.05 with a 95% confidence interval.

## 3. Results

The demographic analysis of the study population is presented in Table 1 and Table 2. A total of 86 white subjects, most of whom lived or had lived in the city of Lodz, Poland, were enumerated and included into the study. There were 25 men (29.1%) and 61 women (70.9%). Our study subjects were divided into three subgroups. Subgroup I included 7 men (35.0%) and 13 women (65.0%). Subgroup II included 6 men (30.0%) and 14 women (70.0%). Subgroup III included 12 men (26.1%) and 34 women (73.9%). Statistical analyses revealed that our three subgroups did not vary significantly in sex (χ^2^ test *p* = 0.760). An analysis of data on the screening visit showed that the mean age of the study subjects was 54.91 ± 8.25 years (range, 29–73 years). The mean BMI value was 25.67 ± 3.86 in the women and 26.45 ± 3.12 in the men (range, 19.26–35.43). The mean GO duration time was 22.59 months ± 23.13 in the women and 19.32 ± 21.55 months in the men (range, 3–12 months). A comparative analysis between subgroup I (latanoprost treatment), subgroup II (therapy with Combigan complex preparation), and subgroup III (observation without any pharmacotherapy) based on the Kruskal–Wallis test revealed that differences in age, BMI value, and GO duration time between the studied subgroups were not statistically significant (*p* = 0.954, *p* = 0.851, and *p* = 0.851, respectively; Table 3 and Table 4). 

The exophthalmos (EXO) secondary to GO ranged from 12.0 to 28.0 mm at the screening visit (Table 5). The mean EXO value in the right eye was 19.18 mm ± 3.58 mm in the women and 21.08 mm ± 2.48 mm in the men. The mean EXO value in the left eye was 19.34 mm ± 3.82 mm in the women and 20.88 mm ± 2.83 mm in the men. The CAS inflammation values ranged from two to six at the screening visit. The mean CAS value in the right eye was 3.54 ± 0.85 in the women and 3.44 ± 0.65 in the men (Table 6). The mean CAS value in the left eye was 3.70 ± 0.78 in the women and 3.56 ± 0.71 in the men. A comparative analysis between subgroup I, subgroup II, and subgroup III based on the Kruskal–Wallis test revealed that the differences in the CAS inflammation values and EXO values between the studied subgroups were not statistically significant either in the right eye (*p* = 0.741 and *p* = 0.279) or in the left eye (*p* = 0.262 and *p* = 0.12). Multiple regression analyses also revealed that the values of exophthalmos were statistically significantly associated with female sex and duration of GO. Female sex decreased exophthalmos by 1.808 and longer duration of GO increased exophthalmos by 0.028 (*p* = 0.002 and *p* = 0.016, respectively). 

The mean IOP parameter in the right eye was 26.72 mmHg ± 1.23 mmHg in the women and 26.32 ± 0.80 mmHg in the men at the screening visit (Table 7). The mean IOP value in the left eye was 26.89 ± 1.93 mmHg in the women and 26.52 ± 1.61 mmHg in the men. A comparative analysis between subgroup I, subgroup II, and subgroup III, based on the Kruskal–Wallis test and Dunn’s test, showed that the differences in the IOP value in the right eye between the studied subgroups on the screening visit were not statistically significant (*p* = 0.450), whereas for the left eye, they were statistically significantly higher in subgroup I and subgroup II compared to subgroup III (*p* = 0.030 and *p* = 0.020, respectively). The difference in the initial IOP in the left eye between subgroup I and subgroup II was not statistically significant (*p* = 1.000).

On the final visit, the mean IOP value in the right eye decreased by 35.7% to 17.20 ± 3.61 mmHg among the patients receiving prostaglandin treatment (subgroup I); in the patients who were administered combined drug Combigan (subgroup II), it decreased by 28.2% to 19.25 ± 2.0 mmHg; and in the control group (subgroup III), not receiving any topical treatment, it decreased by 17.6% to 21.80 ± 4.98 mmHg (Table 8). Similarly, in the left eye, the mean IOP value decreased by 39.1% to 16.70 ± 3.88 mmHg in subgroup I; in subgroup II, it decreased by 28.2% to 19.70 ± 2.41 mmHg; and in subgroup III, it decreased by 17.2% to 21.72 ± 5.11 mmHg. A comparative analysis between subgroup I, subgroup II, and subgroup III based on the Kruskal–Wallis test and Dunn’s test revealed that the differences in IOP value in both eyes on the final visit were statistically significantly lower in subgroup I compared to subgroup III (both *p* = 0.001). The final IOP in subgroup II was also lower than in subgroup III (in both eyes); however, the difference between subgroup II and subgroup III was not statistically significant (*p* = 0.140 and 0.593, respectively). The difference in the final IOP in both eyes between subgroup I and subgroup II were not statistically significant either (*p* = 0.496 and 0.115, respectively); however, the prostaglandin eye drops decreased intraocular pressure more effectively than Combigan drops containing brimonidine and timolol. Topical ocular hypotensive therapy was well tolerated by the patients, both on the check-up appointment and the final visit. 

## 4. Discussion

The previously published studies revealed that neither antithyroid drugs nor thyroidectomy affect the course of GO independent of that caused by the effect of thyroid function [22]. Those studies also revealed that elevated IOP in the course of GO is related to compression of the eyeball by enlarged extraocular muscles, the elevated intraorbital pressure (as result of the proliferation of intraorbital connective tissue), and the enlargement as well as swelling of extraocular muscles. However, orbital decompression and extraocular muscle surgery are effective in lowering the IOP in patients with GO [9,13,14,15,16]; there is a lack of studies concerning topical hypotensive treatment in patients with GO and our study fills this gap. Guminska et al. published a pilot study, which proved that prostaglandin drug latanoprost is effective in lowering IOP in patients with GO on a small group of subjects without a control group in 2014 [17]. The present study, performed within a doctoral thesis, included 172 eyes in 86 individuals, divided into three treatment subgroups including a control group without topical hypotensive treatment. 

One of the inclusion criteria was CAS inflammation score ≥3 in at least one eye, because previously published studies by Behrouzi et al. and Cockerham et al. showed that active GO may result in ocular hypertension and/or progression of glaucoma [7,10]. Due to all included subjects having active GO, all of them received systemic treatment: intravenous corticosteroid therapy at the same cumulative dose of 4.5 g of methylprednisolone, according to the EUGOGO guidelines. The European Thyroid Association and EUGOGO consensus recommended intravenous steroid pulses at an optimal cumulative dose of 4.5–5 g of methylprednisolone (in one course of therapy) as treatment of choice for moderately severe and active GO, but higher doses (up to 8 g) can be used for patients with diplopia and more severe forms [23].

Another important inclusion criterion was IOP ≥ 25 mmHg in both eyes at the screening visit. The results of the Advanced Glaucoma Intervention Study (AGIS) showed that an increase in IOP to 26 mmHg or more increases the risk of glaucoma twelve times in long-term follow-up [24]. The results of Early Manifest Glaucoma Trial (EMGT) showed that a 25% reduction in IOP from the initial values (maintained throughout follow-up) reduced the risk of glaucoma by near 50% [25]. However, the observation period of the patients in our study lasted only 12 weeks; the results of previously published studies showed that prevalence of ocular hypertension in patients with GO is higher than in the general population [7,8,10,11]. Some studies showed that GO is associated with a higher prevalence of open angle glaucoma [7,11], but others showed that prevalence of open-angle glaucoma in patients with GO is similar to that in the general population [8], though their findings were not consistent. 

Spierer and Eisenstein showed that an increase in IOP in patients with active GO correlated positively with the severity of exophthalmos [26] and the results of our study showed female sex decreased exophthalmos by 1.808, and longer duration of GO increased exophthalmos by 0.028 (*p* = 0.002 and *p* = 0.016, respectively) at the screening visit. 

On the final visit, the mean IOP value was significantly lower in all treatment subgroups compared to the initial values. In both subgroups receiving topical treatment, the IOP reduction was higher than in the control group receiving systemic corticosteroids only. Although the difference in the final IOP in both eyes between the prostaglandin (latanoprost) subgroup and the Combigan subgroup was not statistically significant (*p* = 0.496 and 0.115), the latanoprost eye drops decreased intraocular pressure more effectively than Combigan drops containing brimonidine and timolol. Topical ocular hypotensive therapy was well-tolerated by the patients, both on the check-up appointment and the final visit. Our results are in agreement with the results of other previously published studies that showed good ocular-tolerability profile of both types of eye drops [27,28,29]. Although the study by Katz et al. showed fixed-combination brimonidine–timolol was as effective as latanoprost in reducing IOP in patients with glaucoma or ocular hypertension [29], the results of the present study showed better efficacy in lowering IOP of latanoprost versus fix-combination brimonidine–timolol in patients with GO.

The limitations of the current study include the low number of participants, the lack of visual field testing data (which differentiate glaucoma from ocular hypertension), the short period of follow-up, and possible errors when measuring IOP in this group. Patients with GO have increased IOP values on upgaze, so care should be taken during IOP examination [10]. However, it likely had only a minor impact on the study findings. Our study group included only subjects with active Graves’ disease and elevated intraocular pressure equal to or exceeding 25.0 mmHg, and the obtained results are in agreement with the results of other studies from Poland and worldwide. 

## 5. Conclusions

Topical ocular hypotensive treatment in the form of latanoprost (prostaglandin) eye drops or the combined preparation of brimonidine and timolol is effective in reducing intraocular pressure in patients with orbitopathy in the course of Graves’ disease and decreases intraocular pressure more effectively than systemic corticosteroid therapy alone. Latanoprost eye drops lower intraocular pressure more effectively than those containing the combined preparation of brimonidine and timolol in patients with increased intraocular pressure in the course of orbitopathy associated with Graves’ disease. Topical pharmacotherapy is well-tolerated and does not cause any serious side effects. 

## Figures and Tables

**Table 1 ijerph-17-09331-t001:** Analysis of sex of the study population divided into three subgroups.

Sex	Subgroup I		Subgroup II		Subgroup III		All	
	Topical Treatment with Latanoprost and Steroids iv.		Topical Treatment with Brimonidine and Timolol and Steroids iv.		Control Group Steroids iv. Only			
	No	%	No	%	No	%	No	%
Men	7	35.00	6	30.00	12	26.09	25	29.07
Women	13	65.00	14	70.00	34	73.91	61	70.93
All	20	100.00	20	100.00	46	100.00	86	100.00
Statistical analysis	Chi squared test = 0.55, *p* = 0.7603							

**Table 2 ijerph-17-09331-t002:** Analysis of age of the study population divided into three subgroups.

Age [Years]	Subgroup I	Subgroup II	Subgroup III
No of subjects	20	20	46
Minimum	42.00	36.00	29.00
Maximum	67.00	73.00	73.00
Median	56.00	55.00	55.00
Mean	54.95	55.35	54.70
Standard deviation	5.47	10.23	8.46
Asymmetry coefficient	−0.15	−0.26	−0.73
Statistical analysis	Kruskal-Wallis test: H = 0.095; *p* = 0.954		

**Table 3 ijerph-17-09331-t003:** Analysis of BMI values in the study population at the screening visit.

BMI Value	Subgroup I	Subgroup II	Subgroup III
No of subjects	20	20	46
Minimum	19.26	19.82	20.07
Maximum	34.81	34.21	35.43
Median	25.55	25.50	25.19
Mean	26.01	26.33	25.66
Standard deviation	3.55	4.34	3.45
Asymmetry coefficient	0.75	0.50	0.85
Statistical analysis	Kruskal-Wallis test: H = 0.32; *p* = 0.851		

**Table 4 ijerph-17-09331-t004:** Analysis of Graves’ orbitopathy (GO) duration time in the study population at the screening visit.

Duration Time of GO at the Screening Visit (Months)	Subgroup I	Subgroup II	Subgroup III
No of subjects	20	20	46
Minimum (months)	3.00	6.00	3.00
Maximum (months)	72.00	72.00	96.00
Median (months)	8.00	17.00	12.00
Mean (months)	21.95	24.00	20.48
Standard deviation	24.61	21.74	22.50
Asymmetry coefficient	1.24	1.19	1.79
Statistical analysis	Kruskal-Wallis test: H = 1.47; *p* = 0.851		

**Table 5 ijerph-17-09331-t005:** EXO values analysis in the study participant’s right and left eyes at the screening visit.

Exophthalmos Values at the Screening Visit	Right Eyes			Left Eyes		
	Subgroup I	Subgroup II	Subgroup III	Subgroup I	Subgroup II	Subgroup III
No of subjects	20	20	46	20	20	46
Minimum (mm)	12.00	13.00	13.00	14.00	16.00	10.00
Maximum (mm)	22.00	26.00	28.00	24.00	27.00	27.00
Median (mm)	20.00	21.50	20.00	19.00	21.50	19.00
Mean (mm)	19.15	20.65	19.59	19.10	21.20	19.48
Standard deviation	2.81	3.59	3.52	3.11	3.37	3.82
Asymmetry coefficient	−1.10	−0.66	0.07	−0.01	−0.14	0.01
Statistical analysis	Kruskal-Wallis Test: H = 2.55; *p* = 0.279			Kruskal-Wallis Test: H = 4.21; *p* = 0.122		

**Table 6 ijerph-17-09331-t006:** Modified CAS inflammation form analysis in the study participant’s right and left eyes at the screening visit.

Modified CAS Inflammation Values at the Screening Visit	Right Eyes			Left Eyes		
	Subgroup I	Subgroup II	Subgroup III	Subgroup I	Subgroup II	Subgroup III
No of subjects	20	20	46	20	20	46
Minimum	3.00	2.00	2.00	3.00	3.00	3.00
Maximum	6.00	5.00	5.00	6.00	5.00	5.00
Median	3.00	3.00	3.00	3.00	4.00	4.00
Mean	3.45	3.60	3.50	3.50	3.80	3.67
Standard deviation	0.83	0.88	0.75	0.89	0.83	0.67
Asymmetry coefficient	2.05	0.43	0.82	1.75	0.41	0.49
Statistical analysis	Kruskal-Wallis Test: H = 0.60; *p* = 0.741			Kruskal-Wallis Test: H = 2.68; *p* = 0.262		

**Table 7 ijerph-17-09331-t007:** IOP measurements values analysis in the study participant’s right and left eyes at the screening visit.

IOP Values at the Screening Visit (mmHg)	Right Eyes			Left Eyes		
	Subgroup I	Subgroup II	Subgroup III	Subgroup I	Subgroup II	Subgroup III
No of subjects	20	20	46	20	20	46
Minimum (mmHg)	25.00	25.00	25.00	25.00	25.00	25.00
Maximum (mmHg)	29.00	29.00	29.00	35.00	34.00	29.00
Median (mmHg)	26.00	26.50	26.00	27.00	27.00	26.00
Mean (mmHg)	26.75	26.80	26.46	27.40	27.45	26.22
Standard deviation	1.25	1.24	1.03	2.21	2.04	1.38
Asymmetry coefficient	0.71	0.60	1.28	2.33	1.84	1.02
Statistical analysis	Kruskal-Wallis Test: H = 1.60; *p* = 0.450			Kruskal-Wallis Test: H = 11.59; *p* = 0.0030		
				Dunn’s test		
Subgroup I					0.127	2.581
Subgroup II				0.127		2.730
Subgroup III				2.581	2.730	
				*p*		
Subgroup I					1.000000	0.029565
Subgroup II				1.000000		0.018978
Subgroup III				0.029565	0.018978	

**Table 8 ijerph-17-09331-t008:** IOP measurements values analysis in the study participant’s right and left eyes at the final visit.

IOP Values at the Final Visit (mmHg)	Right Eyes			Left Eyes		
	Subgroup I	Subgroup II	Subgroup III	Subgroup I	Subgroup II	Subgroup III
No of subjects	20	20	46	20	20	46
Minimum (mmHg)	10.00	15.00	12.00	10.00	15.00	12.00
Maximum (mmHg)	23.00	22.00	30.00	23.00	24.00	32.00
Median (mmHg)	18.00	20.00	22.00	17.50	20.00	21.50
Mean (mmHg)	17.20	19.25	21.80	16.70	19.70	21.72
Standard deviation	3.61	2.00	4.98	3.88	2.41	5.11
Asymmetry coefficient	−0.58	−0.56	−0.23	−0.17	−0.51	0.05
Statistical analysis	Kruskal-Wallis Test: H = 21.05; *p* = 0.0008			Kruskal-Wallis Test: H = 14.11; *p* = 0.0009		
	Dunn’s test			Dunn’s test		
Subgroup I		1.387	3.642		2.074	3.737
Subgroup II	1.387		2.005	2.074		1.288
Subgroup III	3.642	2.005		3.737	1.288	
	*p*			*p*		
Subgroup I		0.496561	0.000812		0.114299	0.000560
Subgroup II	0.496561		0.134977	0.114299		0.593070
Subgroup III	0.000812	0.134977		0.000560	0.593070	

## Data Availability

The source data is available at Department of Ophthalmology and Visual Rehabilitation, Central Veterans Hospital, Medical University of Lodz, 113 Zeromskiego str., Lodz, Poland.

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
