# Peer review of "Topical Treatment of Elevated Intraocular Pressure in Patients with Graves’ Orbitopathy"

_ijerph, 2020, doi:10.3390/ijerph17249331_

Round 1

Reviewer 1 Report

This manuscript is aimed to evaluate the efficacy of topical hypotensive treatment and/or systemic corticosteroids therapy in patients with elevated intraocular pressure and Graves’ orbitopathy (GO). Authors concluded that topical ocular hypotensive treatment is effective in reducing intraocular pressure in 30 GO and decreases intraocular pressure more effectively than systemic corticosteroid therapy alone. Some questions are concerned before consideration of publications.

Major concerns:

1.Though the main issue in this manuscript is the ocular hypertension and GO, however, the other glaucoma test such as visual field is lacking, the percentage of real glaucoma in this study is lacking. Therefore, the main issue is should we treat ocular hypertension in every GO patients with elevation of IOP? Especially intravenous steroid therapy also can decrease the intraocular pressure without anti-glaucoma eyedrops. Therefore, the value of this study is lessened as only probing the IOP in a short term.

2. How is the intraocular pressure measured? The position of eyeball may affect the intraocular pressure (

Journal of Glaucoma:  2008 -  17 -  p 249-250 doi: 10.1097/IJG.0b013e31816c4db3)

3. In this prospective study, the study subjects  were divided into three treatment subgroups. How is the method to divide patients into groups? Is it randomized or selected?

4. The observation period of the patients in our study lasted only 12 weeks, that is too short to observe whether the ocular hypertension will develop into glaucoma or not.

5. Ethics concerns: In group three, if patients had evidence of glaucoma (visual field test is lacking in ophthalmic examinations), it is not appropriate to put patients only in steroid treatment

Author Response

Dear Editors, Dear Reviewers,

 We would like to thank you and other reviewers for your kind, friendly and instructive comments on our paper. Following your suggestions we have made some corrections. Please find my responses and list of the changes along with modified manuscript as an attached file. 

Yours sincerely

            Michal S. Nowak MD, PhD 

Reviewer 1

This manuscript is aimed to evaluate the efficacy of topical hypotensive treatment and/or systemic corticosteroids therapy in patients with elevated intraocular pressure and Graves’ orbitopathy (GO). Authors concluded that topical ocular hypotensive treatment is effective in reducing intraocular pressure in 30 GO and decreases intraocular pressure more effectively than systemic corticosteroid therapy alone. Some questions are concerned before consideration of publications.

Major concerns:

1.Though the main issue in this manuscript is the ocular hypertension and GO, however, the other glaucoma test such as visual field is lacking, the percentage of real glaucoma in this study is lacking. Therefore, the main issue is should we treat ocular hypertension in every GO patients with elevation of IOP? Especially intravenous steroid therapy also can decrease the intraocular pressure without anti-glaucoma eyedrops. Therefore, the value of this study is lessened as only probing the IOP in a short term.

Ad. 1 Yes, we agree with this comment. It is estimated that elevated intraocular pressure (IOP) in the course of Graves’ orbitopathy affects 3.7%-24% of patients.  Elevated IOP present in GO is a very significant sign, not included in the disease severity classification. In consequence, it may result in glaucomatous optic neuropathy, because (according to the European Glaucoma Society guidelines) ocular hypertension is the most important risk factor for the development of glaucoma According to various authors, glaucoma may occur in the course of GO in 0.8 to 13% of patients. This information was stated in modified manuscript: please see the introduction section, page 1-2, lines 37-38, 42-44 and 46-47.

We have added into the introduction section the new paragraphs, which described the mechanism of IOP elevation in GO patients as well as drugs pathways:

“The IOP is determined by aqueous humour production, aqueous humour outflow and the level of episcleral venous pressure. In GO increased orbital pressure results in increased episcleral venous pressure leading to elevated IOP.” Page 2, lines 44-46

“Prostaglandin-group drug (prostaglandin F2α analogue- latanoprost) and a complex drug being a combination of brimonidine (clonidine derivative, α2-selective agonist) and timolol (nonselective β receptor antagonist) were selected for our study. Prostaglandin-group drugs (i.e. latanoprost) increase the outflow of aqueous humour and a complex drug being a combination of brimonidine and timolol (i.e. Combigan) decreases the production and simultaneously increases the outflow of aqueous humour. Latanoprost and Combigan showed good efficacy in decreasing IOP and good ocular-tolerability profile in patients with glaucoma or ocular hypertension in many clinical studies. The systemic corticosteroids therapy decreases the activity of the tissue inflammation in GO patients in both genomic and non-genomic pathways. However, patients being treated with systemic corticosteroids should first be screened for liver dysfunction, uncontrolled hypertension and/or diabetes, history of peptic ulcer, and glaucoma, and then monitored for possible adverse events.” Page 2, lines 54-65.

We have also added new references to the references section:

  • Bahn RS. The Graves’ ophthalmopathy. N Engl J Med. 2010; 362: 726-238
  • Wiersinga WM, Kahaly GJ. Graves’ Orbitopathy: a multidisciplinary approach – questions and answers. Karger, Basel 2017: 128-155
  • Bartelana L, Baldeschi L, Boboridis K, Eckstein A, Kahaly GJ, Marcocci C, Perros P, Salvi M, Wiersinga WM. The 2016 European Thyroid Association/European Group on Graves’ Orbitopathy guidelines for the management of Grves’ orbitopathy. Eur Thyroid J 2016; 5: 9-26

The history of topical hypotensive treatment prior to the study and closed or narrow angle glaucoma was the exclusion criterion for our study. This information was stated in modified manuscript. Please see: page 2, lines 87-88.

However, the observation period of the patients in our study lasted only 12 weeks, the results of previously published studies showed that prevalence of ocular hypertension in patients with GO is higher than in the general population. This information was stated in modified manuscript. Please see: page 8, lines 210-212.

  1. How is the intraocular pressure measured? The position of eyeball may affect the intraocular pressure. (Journal of Glaucoma:  2008 -  17 -  p 249-250 doi: 10.1097/IJG.0b013e31816c4db3)

Ad. 2 Yes, we agree with this comment. Intraocular pressure (IOP) was measured with the Goldman applanation tonometry. This information was stated in modified manuscript. Please see: page 2, lines 81-82.

We have also added into the discussion section the new limitation: “The limitation of the current study include low number of participants, short period of follow-up and possible errors when measuring IOP in this group. Patients with GO have increased IOP values on upgaze, so great care should be taken during IOP examination.” Please see: page 8, lines 233-235.

  1. In this prospective study, the study subjects  were divided into three treatment subgroups. How is the method to divide patients into groups? Is it randomized or selected?

Ad. 3 Yes, we agree with this comment. We changed the information in the materials and methods section: “The study subjects were randomly selected into three treatment subgroups.”

Please see: page 2, lines 91-92.

  1. The observation period of the patients in our study lasted only 12 weeks, that is too short to observe whether the ocular hypertension will develop into glaucoma or not.

Ad. 4 Yes, we agree with this comment. However, the observation period of the patients in our study lasted only 12 weeks, the results of previously published studies showed that prevalence of ocular hypertension in patients with GO is higher than in the general population. This information was stated in modified manuscript. Please see: page 8, lines 210-212.

  1. Ethics concerns: In group three, if patients had evidence of glaucoma (visual field test is lacking in ophthalmic examinations), it is not appropriate to put patients only in steroid treatment

Ad. 5 Yes, we agree with this comment. The history of topical hypotensive treatment prior to the study and closed or narrow angle glaucoma was the exclusion criterion for our study. This information was stated in modified manuscript. Please see: page 2, lines 87-88.

We have also made some other changes according to others Reviewers suggestions and uploaded the modified manuscript. Please see the attachments.

Reviewer 2 Report

Comments:

  1. It is well known that women affected Graves disease (hyperthrodism, exophthalmic goiter, goiter) more than men, with a ratio ranging from 5:1 to 7:1, and more common between the ages 20 and 50 (Goodman and Gilman’s the pharmacological basis of therapeutics, 13ed, page 793).
  2. In this manuscript thyroid-associated orbitopathy (Graves’ orbitopathy, GO) was briefly introduced. Patients with GO, 25 males (24.1%) and 61 females (70.9%) whom live or have lived in the city of Lodz were divided into 3 subgroups, subgroup Ⅰ local treated with latanoprost (prostaglandin F2α analogue GD, subgroup Ⅱ local treated with brimonidine (clonidine derivative, α2-selective agonist) and timolol (nonselective β receptor antagonist) preparation BID, and subgroup Ⅲ as control group, all three subgroups received systemic methylprednisolone 0.5mg weekly for the first six weeks, than 0.25mg of same drug for the next six weeks.
  3. The study was well designed the collected data were suitable to publish in this Journal.

Suggestions:

  1. “‘Graves’ ophthalmopathy” was used in line 3 and line 125, and “Graves’ orbitopathy” was used in all other places in this manuscript. It is suggested to unify then.
  2. Removing part or all the thyroid gland, inhibition of thyroid hormone synthesis or release, are the principle for the treatment of Graves disease. Please discussed are GO should be treat with same methods.
  3. Please describe the classification of drugs used in subgroup Ⅰ and Ⅱ, in Comment 2.
  4. Please indicated the concentration of Latamoprest used in subgroup Ⅰ, Brimonidine and Timolol in combigan in subgroup Ⅱ.
  5. Half-life of methylprednisolone is 2.3±0.5 hours. Please discussed the doses used in this study.
  6. The presentation style of References list should be rechecked.

Author Response

Dear Editors, Dear Reviewers,

 We would like to thank you and other reviewers for your kind, friendly and instructive comments on our paper. Following your suggestions we have made some corrections. Please find my responses and list of the changes along with modified manuscript as an attached file. 

Yours sincerely

            Michal S. Nowak MD, PhD 

Reviewer 2

  • It is well known that women affected Graves disease (hyperthrodism, exophthalmic goiter, goiter) more than men, with a ratio ranging from 5:1 to 7:1, and more common between the ages 20 and 50 (Goodman and Gilman’s the pharmacological basis of therapeutics, 13ed, page 793).
  • In this manuscript thyroid-associated orbitopathy (Graves’ orbitopathy, GO) was briefly introduced. Patients with GO, 25 males (24.1%) and 61 females (70.9%) whom live or have lived in the city of Lodz were divided into 3 subgroups, subgroup local treated with latanoprost (prostaglandin F2α analogue GD, subgroup local treated with brimonidine (clonidine derivative, α2-selective agonist) and timolol (nonselective β receptor antagonist) preparation BID, and subgroup as control group, all three subgroups received systemic methylprednisolone 0.5mg weekly for the first six weeks, than 0.25mg of same drug for the next six weeks.
  • The study was well designed the collected data were suitable to publish in this Journal.

Suggestions:

  1. “‘Graves’ ophthalmopathy” was used in line 3 and line 125, and “Graves’ orbitopathy” was used in all other places in this manuscript. It is suggested to unify then.

Ad. 1 Yes, we agree with this comment. We have unified the terminology into Graves’ orbitopathy (GO), we have changed the title of the manuscript and changed the text in the whole manuscript. Please see the attachments.

  1. Removing part or all the thyroid gland, inhibition of thyroid hormone synthesis or release, are the principle for the treatment of Graves disease. Please discussed are GO should be treat with same methods.

Ad. 2 Yes, we agree with this comment. The previously published studies revealed that neither antithyroid drugs nor thyroidectomy affect the course of GO, independently of that caused by the effect of thyroid function. This information was added into the discussion section. Please see: page 7, lines 184-185.

We have also added new references to the references section:

  • Bahn RS. The Graves’ ophthalmopathy. N Engl J Med. 2010; 362: 726-238
  • Wiersinga WM, Kahaly GJ. Graves’ Orbitopathy: a multidisciplinary approach – questions and answers. Karger, Basel 2017: 128-155
  • Bartelana L, Baldeschi L, Boboridis K, Eckstein A, Kahaly GJ, Marcocci C, Perros P, Salvi M, Wiersinga WM. The 2016 European Thyroid Association/European Group on Graves’ Orbitopathy guidelines for the management of Grves’ orbitopathy. Eur Thyroid J 2016; 5: 9-26

  1. Please describe the classification of drugs used in subgroup and , in Comment 2.

Ad. 3 Yes, we agree with this comment. We have described the classification of drugs used in subgroup I and II in the modified introduction section: “Prostaglandin-group drug (prostaglandin F2α analogue- latanoprost) and a complex drug being a combination of brimonidine (clonidine derivative, α2-selective agonist) and timolol (nonselective β receptor antagonist) were selected for our study.” Page 2, lines 54-56.

  1. Please indicated the concentration of Latamoprest used in subgroup , Brimonidine and Timolol in combigan in subgroup .

Ad. 4 Yes, we agree with this comment. We have added the concentrations of latanoprost, brimonidine and timolol in the modified materials and methods section. Please see: page 2 , line 94 and page 3 , line 96.

  1. Half-life of methylprednisolone is 2.3±0.5 hours. Please discussed the doses used in this study.

Ad. 5 Yes, we agree with this comment. We have discussed it in the modified manuscript: “The European Thyroid Association and EUGOGO consensus recommended intravenous steroid pulses at optimal cumulative dose of 4.5 g – 5 g of methylprednisolone (in one course of therapy) as treatment of choice for moderately severe and active GO, but higher doses (up to 8 g) can be used for patients with diplopia and more severe forms.” Please see: page 8, lines 201-204

  1. The presentation style of References list should be rechecked.

Ad. 6 Yes, we agree with this comment. The presentation style of references section was rechecked. Please see in modified manuscript.

We have also made some other changes according to others Reviewers suggestions and uploaded the modified manuscript. Please see the attachments.

Reviewer 3 Report

Please changes in all tables:

Maksimum to Maximum

coeffitient to coefficient

participants’ to participant’s

line 146: OIP to IOP

Table7: , Test Dunna = Test Dunn’s, Dunn’s test is the correct term

Other corrections:

  • Explain the drug pathways in the disease condition in the introduction 
  • Explain the genetic factors (underlying) that may/may not involve in the medication. Did the patients had underlying genetic disorders?
  • Does the control group was treated as placebo/saline?
  •  

Author Response

Dear Editors, Dear Reviewers,

 We would like to thank you and other reviewers for your kind, friendly and instructive comments on our paper. Following your suggestions we have made some corrections. Please find my responses and list of the changes along with modified manuscript as an attached file. 

Yours sincerely

            Michal S. Nowak MD, PhD 

Reviewer 3

  1. Please changes in all tables:

Maksimum to Maximum

coeffitient to coefficient

participants’ to participant’s

line 146: OIP to IOP

Table7: , Test Dunna = Test Dunn’s, Dunn’s test is the correct term

Ad 1. Yes, we agree with this comment. We have corrected all tables included in the manuscript. Please see the attachments.

  1. Other corrections:
  2. Explain the drug pathways in the disease condition in the introduction 

Ad. a Yes, we agree with this comment. We have added into the introduction section the new paragraphs, which described the mechanism of IOP elevation in GO patients as well as drugs pathways:

“The IOP is determined by aqueous humour production, aqueous humour outflow and the level of episcleral venous pressure. In GO increased orbital pressure results in increased episcleral venous pressure leading to elevated IOP.” Page 2, lines 44-46

“Prostaglandin-group drug (prostaglandin F2α analogue- latanoprost) and a complex drug being a combination of brimonidine (clonidine derivative, α2-selective agonist) and timolol (nonselective β receptor antagonist) were selected for our study. Prostaglandin-group drugs (i.e. latanoprost) increase the outflow of aqueous humour and a complex drug being a combination of brimonidine and timolol (i.e. Combigan) decreases the production and simultaneously increases the outflow of aqueous humour. Latanoprost and Combigan showed good efficacy in decreasing IOP and good ocular-tolerability profile in patients with glaucoma or ocular hypertension in many clinical studies. The systemic corticosteroids therapy decreases the activity of the tissue inflammation in GO patients in both genomic and non-genomic pathways. However, patients being treated with systemic corticosteroids should first be screened for liver dysfunction, uncontrolled hypertension and/or diabetes, history of peptic ulcer, and glaucoma, and then monitored for possible adverse events.” Page 2, lines 54-65.

We have also added new references to the references section:

  • Bahn RS. The Graves’ ophthalmopathy. N Engl J Med. 2010; 362: 726-238
  • Wiersinga WM, Kahaly GJ. Graves’ Orbitopathy: a multidisciplinary approach – questions and answers. Karger, Basel 2017: 128-155
  • Bartelana L, Baldeschi L, Boboridis K, Eckstein A, Kahaly GJ, Marcocci C, Perros P, Salvi M, Wiersinga WM. The 2016 European Thyroid Association/European Group on Graves’ Orbitopathy guidelines for the management of Grves’ orbitopathy. Eur Thyroid J 2016; 5: 9-26

  1. Explain the genetic factors (underlying) that may/may not involve in the medication. Did the patients had underlying genetic disorders?

Ad. b We apologize but we do not know if we fully understood which genetic factors may affect the treatment of Graves’ orbitopathy. Both genetic and environmental factors may increase the risk to develop GO in patients with Graves’ disease. Specifically, advanced age, male sex, tobacco use, biochemically more severe hyperthyroidism and high TSH receptor antibodies have been identified as risk factors, as well as 131 I therapy. However, we did not screened our study population for genetic disorders, our study showed female gender decreased exophthalmos by 1.808 and longer duration of GO increased exophthalmos by 0.028 (p= 0.002 and p=0.016 respectively), at the screening visit.

This information was stated in modified manuscript. Please see: page 8, lines 217-219.

It is also well known that systemic corticosteroids therapy decreases the activity of the tissue inflammation in GO patients in both genomic and non-genomic pathways. However, patients being treated with systemic corticosteroids should first be screened for liver disfunction, uncontrolled hypertension and/or diabetes, history of peptic ulcer, and glaucoma, and then monitored for possible adverse events. We have added this information into the modified introduction as stated earlier. Page 2, lines 54-65.

  1. Does the control group was treated as placebo/saline?

Ad. c Yes, we agree with this comment. We have added this information into the materials and methods section: “Subgroup III – 46 patients (12 males and 34 females) with GO and intraocular pressure equal to or exceeding 25.0 mmHg were qualified for the control group, neither received topical hypotensive treatment nor placebo eye drops.” Please see: page 3, lines 97-99

We have also made some other changes according to others Reviewers suggestions and uploaded the modified manuscript. Please see the attachments.

Round 2

Reviewer 1 Report

1.Though authors have answered some raised questions, but the visual field testing data in all enrolled patients is still lacking.

2. In clinic practice, in patients with ocular hypertension, the exact management is to differentiate whether it is a glaucoma or a ocular hypertention first by visual field testing, optic coherence tomography of thickness of nerve fiber layer, etc. Not starting ocular hypotensive drugs just based upon elevated intraocular pressure. Therefore, this study design is flawed.

Author Response

Dear Editors, Dear Reviewers,

 We would like to thank you and other reviewers for your kind, friendly and instructive comments on our paper. Following your suggestions we have made some corrections. Please find my responses and list of the changes along with modified manuscript as an attached file. 

Yours sincerely

            Michal S. Nowak MD, PhD 

Reviewer 1

  1. Though authors have answered some raised questions, but the visual field testing data in all enrolled patients is still lacking.

Ad. 1 Yes, we agree with this comment. However, the visual field testing data in all enrolled patients is still lacking, a history of topical hypotensive treatment prior to the study, closed or narrow angle glaucoma was an exclusion criterion for the present study. This information was already stated in the modified manuscript. Please see: page 2, lines 87-88.

We have also added into the discussion section the new limitation: “The limitations of the current study include low number of participants, lack of visual field testing data (which differentiate glaucoma from ocular hypertension), short period of follow-up and possible errors when measuring IOP in this group.”  Please see: page 8, lines 233-235.

  1. In clinic practice, in patients with ocular hypertension, the exact management is to differentiate whether it is a glaucoma or a ocular hypertention first by visual field testing, optic coherence tomography of thickness of nerve fiber layer, etc. Not starting ocular hypotensive drugs just based upon elevated intraocular pressure. Therefore, this study design is flawed.

Ad. 2 We apologize but we do not know if we fully understood why the study design is flawed. Elevated IOP present in GO is a very significant sign, not included in the disease severity classification. In consequence, it may result in glaucomatous optic neuropathy, because (according to the European Glaucoma Society guidelines) ocular hypertension is the most important risk factor for the development of glaucoma. The results of previously published studies showed that prevalence of ocular hypertension and glaucoma in patients with GO is higher than in the general population. It is estimated that elevated intraocular pressure (IOP) in the course of GO affects 3.7%-24% of patients and glaucoma may occur in the course of GO in 0.8 to 13% of patients. This information was already stated in the modified manuscript: please see the introduction section, page 1-2, lines 37-38 and 46-47.

The aim of the present study was the evaluation of the efficacy of elevated IOP treatment with the use of topical therapy in patients with orbitopathy in the course of Graves’ disease. A history of topical hypotensive treatment prior to the study, closed or narrow angle glaucoma was an exclusion criterion for the present study. This information was already stated in the modified manuscript. Please see: page 2, lines 87-88.

The aim of our study was not to differentiate glaucoma from ocular hypertension in our study group because the observation period of the patients lasted only 12 weeks.  

Round 3

Reviewer 1 Report

The following papers already showed the glaucoma evidence was similar to the general population.1,2 So, the treatment of glaucoma or ocular hypertension in patients with GO should be the same with the general populations.

References  1.  The prevalence of OHT in patients with GO was higher and the prevalence of OAG was similar to that in the general population. ( J Glaucoma 2018 May;27(5):464-469.) 2. The present study did not reveal a statistically significant difference in the prevalence of ocular hypertension or glaucoma between patients with Graves' orbitopathy and the general population. (Eye 2009 Apr;23(4):957-9.)